# An Interactive Web Application for Decision Tree Learning

**Miriam Elia** [1]   **Carola Gajek** [1]   **Alexander Schiendorfer** [1]   **Wolfgang Reif** [1]

## Abstract

Decision tree learning offers an intuitive and straightforward introduction to machine learning techniques, especially when students are used to program imperative code. Most commonly, trees are trained using a greedy algorithm based on information-theoretic criteria. While there are many static resources such as slides or animations out there, interactive visualizations tend to be based on somewhat outdated UI technology and dense in information. We propose a clean and simple web application for decision tree learning that is extensible and open source.[1]

## 1. Decision Tree Learning

Decision tree learning is commonly studied among the first Machine Learning models in an introductory level course. The range of applications for decision trees (DT), in particular as ensemble methods such as boosted trees or random forests, nowadays is enormous. Searching for "decision tree application contexts" in Google scholar, for instance, results in hundreds of papers published after 2019 using or analyzing DTs. Domains reach from autonomous driving (Alam et al., 2020), over biogas plants (Vondra et al., 2019) to politics (Kaufman et al., 2019). But why is it that such a conceptually simple model, e.g. not as complex as neural networks, is so widely used and thus relevant for any student of Machine Learning (ML)? The answer is also simple: they provide impeccable results when used on data sets, allow for diverse tasks, i.e. regression, classification and multi-output (Géron, 2017), and last but not least, DTs build the foundation of random forests (RF) which are according to Géron "among the most powerful Machine Learning

algorithms available today" (Géron, 2017). More recently, decision trees have experienced a surge in interest due to the demand for explainable AI (Lundberg et al., 2020). Therefore, it is very important for anyone interested in ML to be well-versed with DTs. This paper presents a web application (see Figure 1) that has been designed to serve exactly that purpose: teaching DTs to beginners.

## 2. How to Train Decision Trees

Since finding optimal DTs on a given training set is known to be NP-complete (Mitchell, 1997), most often these trees are grown in a greedy fashion, going back to the ID3 algorithm developed in 1979 by Quinlan (Quinlan, 1986). The basic procedure is to reduce entropy (or another purity-related criterion) as much as possible from the root node to the leaf nodes of the DT. The data set comprises a number of instances which respectively consist of concrete values related to a number of attributes (or features) and a label or class value which is the result that the DT aims to classify correctly. The classical instructional example is the so-called "tennis data set". It comprises 14 instances, the attributes {Outlook, Temperature, Humidity, Windy} and the label values {N, P}, for negative and positive, i.e. if the person plays tennis under these weather conditions or not. This data set (see Table 1) is also used in the screenshots of the web application.

In his book "Machine Learning", Mitchell (Mitchell, 1997) illustrates the algorithm to create a DT. In the first step, the entropy of the root node is calculated. The algorithm then requires the calculation of all possible posterior entropy values obtained after splitting according to every attribute. Thereafter, the information gain is calculated by considering the difference between the root node's entropy and the weighted sum of the resulting child nodes' entropy values. This step is repeated for every present attribute, and the attribute that generates the highest information gain is chosen. This greedy procedure is repeated until all nodes are *pure*, i.e., have an entropy of 0.[2]

---

[1]Institute for Software & Systems Engineering, University of Augsburg, Augsburg, Germany. Correspondence to: Alexander Schiendorfer <schiendorfer@isse.de>, Miriam Elia <miriam@elianet.de>.

*Proceedings of the 35th International Conference on Machine Learning*, Stockholm, Sweden, PMLR 80, 2018. Copyright 2018 by the author(s).

[1]Source available under https://github.com/isse-augsburg/decision-tree-learning-ecml.

---

[2]This is the ideal case. If the tree depth is constrained and there exist leaf nodes with an entropy higher than 0, according to the algorithm, the label of such a node refers to the most common label value of the node's data points (Mitchell, 1997).

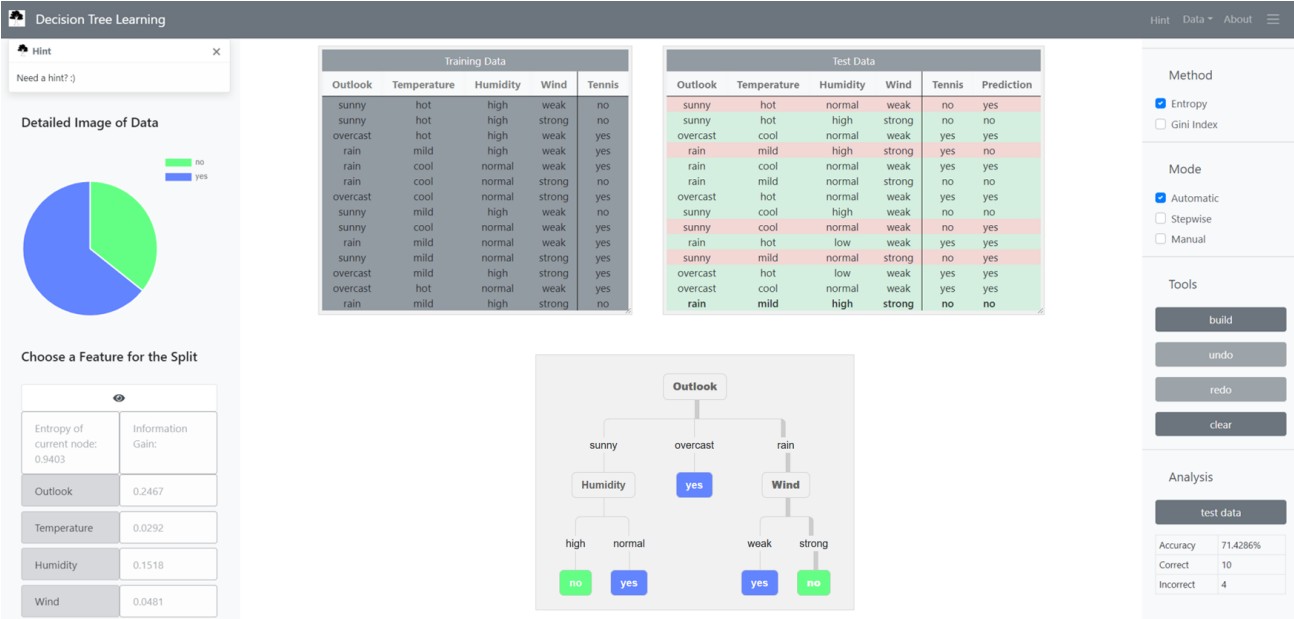

*Figure 1.* A complete view of the web application: On the left side, detailed information of the root node is displayed. The test data is tested on the complete DT, and the path used by the DT to classify the selected test instance, i.e. the last one of the test data <"rain", "mild", "high", "strong", "no">, is highlighted.

*Table 1.* The tennis data set with labels "yes" and "no" instead of originally "positive" and "negative".

| OUTLOOK | TEMP | HUMIDITY | WIND | TENNIS? |
|---------|------|----------|------|---------|
| SUNNY | HOT | HIGH | WEAK | NO |
| SUNNY | HOT | HIGH | STRONG | NO |
| OVERCAST | HOT | HIGH | WEAK | YES |
| RAIN | MILD | HIGH | WEAK | YES |
| RAIN | COOL | NORMAL | WEAK | YES |
| RAIN | COOL | NORMAL | STRONG | NO |
| OVERCAST | COOL | NORMAL | STRONG | YES |
| SUNNY | MILD | HIGH | WEAK | NO |
| SUNNY | COOL | NORMAL | WEAK | YES |
| RAIN | MILD | NORMAL | WEAK | YES |
| SUNNY | MILD | NORMAL | STRONG | YES |
| OVERCAST | MILD | HIGH | STRONG | YES |
| OVERCAST | HOT | NORMAL | WEAK | YES |
| RAIN | MILD | HIGH | STRONG | NO |

## 3. A Web Application for Interactive DT Learning

Our tool was inspired by AI Space's[3] software to learn DTs, and offers a first contact to algorithms that learn rules from data by creating an inductive model and is thus well-suited for introducing ML. Conventional programming usually leaves it up to the programmer to tell the program what to

do, a classical example would be a collection of `if`-clauses. DTs however, offer an understandable introduction to the "new way of thinking" inherent to machine learning (Géron, 2017). The goal of the web application is to provide enough room for experimentation to keep learners engaged, so that they both internalize the general mindset of ML and gain a thorough understanding of DTs. Yet, it strives to focus on the essential parts of the algorithm in order to avoid overwhelming first-time learners. Furthermore, as web application, the software is easy to distribute and maintain and it is open source and free to use for everybody. Its design aims to grant an pleasant user experience through its modern style and self-explanatory structure. Supporting usability friendliness, the logical entities in the center of the tool, i.e. the tables containing the imported training and test data and the DT, can be dragged and modified in size (not the DT) according to the user's wishes. The setting on the right side of the tool is implemented as an expandable sidebar which can be hidden if the user needs more space to build the DT.

### 3.1. The Target Audience

The target audience of the web application are beginners in machine learning who tend to have a background in computer science. This could be university students, seminar participants of a company that aims to introduce AI and ML, or even pupils at school. The basic setup of a typical supervised learning context is illustrated by the web

---

[3] http://www.aispace.org/dTree/index.shtml

app: the data is separated into training and test data while the model learns on the training set and its performance is evaluated on the test set. When the user clicks on an already classified test instance, the path determined by the algorithm to classify the instance is displayed in the tree, as shown in Figure 1. Further, the instances in the test set are highlighted *green* if predicted correctly, and *red* otherwise. The predicted class is shown on the right side of the test set. If no matching leaf node could be found, "none" is listed. Concerning the possible data types, the tool will work with data sets consisting of numeric values only, both numeric values and categorical values or categorical values only.

### 3.1.1. THE STUDENTS' PERSPECTIVE

In addition to internalizing basic concepts, the student will gain a thorough understanding of DTs and how they work, following the greedy decision tree learning algorithm. The web application offers three different modes, as shown in Figure 4: the *automatic* mode builds a DT purity following ID3 either with the Shannon entropy or Gini index as split metric. The *stepwise* mode is more relevant for the teacher and allows to step through the splitting decisions. But the third, *manual* mode is the application's key feature and crucial for the student's learning process. In manual mode, the web application enables the user to play around with trees of different depths. By comparing the performance of those DTs with the performance of DTs built by ID3, students obtain a basic feeling for the relation between the information gain and the entropy of the nodes. When students click on the node they would like to split next, as shown in Figure 2, the calculated information gain is listed for every feature, respectively. Further, the node's distribution is illustrated in a pie chart, while every target class is referred to by a different color (10 is the maximally acceptable total number of different labels). This deepens the understanding of the concept of entropy: If students follow the algorithm and choose the splits that generate the maximum gain, fewer colors, and thus less "chaos", are shown when clicking on a node. The final nodes contain just one single color, i.e. have the minimal entropy of 0. If students choose a smaller gain, it will generally take much longer to achieve pure leaf nodes.

By experimenting with different trees, testing them, and evaluating their accuracy, concepts like overfitting can be observed. To illustrate this, the tool offers to analyze the performance of DTs by displaying the number of correctly and incorrectly classified instances, and the calculated accuracy of evaluating the model on the test set as shown in Figure 4. Overfitting manifests itself by having a high training accuracy and low testing accuracy. Students see this if the learned DTs focus on spurious correlations that were only observed in the training data. Underfitting occurs when the model is too simple to learn the underlying structure of the

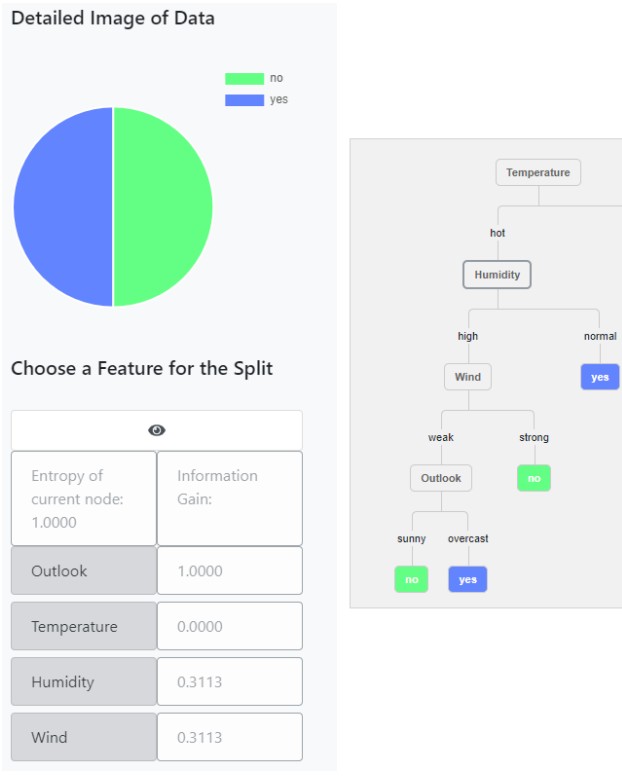

*Figure 2.* Detailed information of the clicked node "Humidity": the distribution of the corresponding instances' target labels is displayed in the pie chart. The table underneath shows the information gain of all next split choices, i.e. the four features. For example, splitting according to "Temperature" would not discriminate the node's data set any further.

data (Géron, 2017). In this case, the DT cannot even classify the training set correctly. An underfitting tree is harder to obtain using our tool since it trains to purity. Constraining the trees' depths or preparing suitable data sets mitigates this issue.

To help students come up with ideas (e.g., the next greedy split proposal), a dialog box with hints appears on the left upper corner of the web application. If the mode is not set to manual, the content changes to theoretical information. Exemplifying hints are illustrated in Figure 3.

### 3.1.2. THE TEACHER'S PERSPECTIVE

Teachers have to select the training and test sets that should be used. The default data set so far is a data set specifying monsters, where input features like color, food, or speed are mapped to a class of monsters, for instance vampires or werewolves. The final DTs can also be saved and loaded into the tool which enables teachers to prepare a short introduction for explaining ID3 and how different DTs can

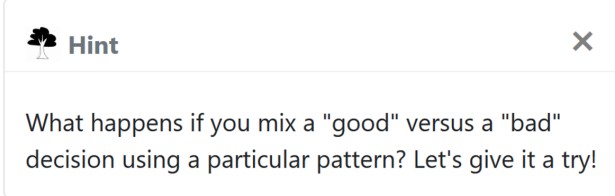

*Figure 3.* An example for hint messages displayed in the web application appearing in manual mode.

be created. The stepwise mode can be used to explain the calculated values for the entropy or Gini index of the current node and the information gain more profoundly. Further, as already mentioned above, teachers can prepare a specific tree to explain concepts like overfitting. To explain underfitting, other models such as linear regression might currently be better suited. To ease the preparation of sessions for the teacher, the tool offers the option to save and load complete sessions in or from a text file, which include the training and test data, the mode, and the split attributes according to which the tree has been built, as illustrated in Figure 4. Saving sessions is only possible in manual mode, as the automatically created DT needs simply one click to be build.

### 3.2. Usage and Customization

It is possible to upload individually generated data sets by complying to the data format, i.e. separating each value in the CSV file by a semi-colon, and the tool interprets the first row as attributes and the last column as label values. By customizing the data set, the teacher can on the one hand adapt to diverse target groups, and on the other hand, convey different content. The "monster" set for instance, is fun and aims to support students and pupils in playfully learning the theory. Another data set example could contain "smoking", "fast food consumption", and "physical fitness" as features and aim to classify the instances, i.e. people, according to health insurance suitability. This could raise awareness of the ethical responsibility when using ML tools.

We explicitly **invite the online community** to share and extend these custom or customized data sets to enhance the value of the web app even further since data sets dedicated to specific algorithmic or ethical issues are still scarce.

### 4. Conclusion and Future Work

We presented a web application that demonstrates greedy decision tree learning based on the ID3 algorithm in manual, stepwise, and automatic mode. Using modern UI concepts and visualizations such as pie charts, we hope to provide a

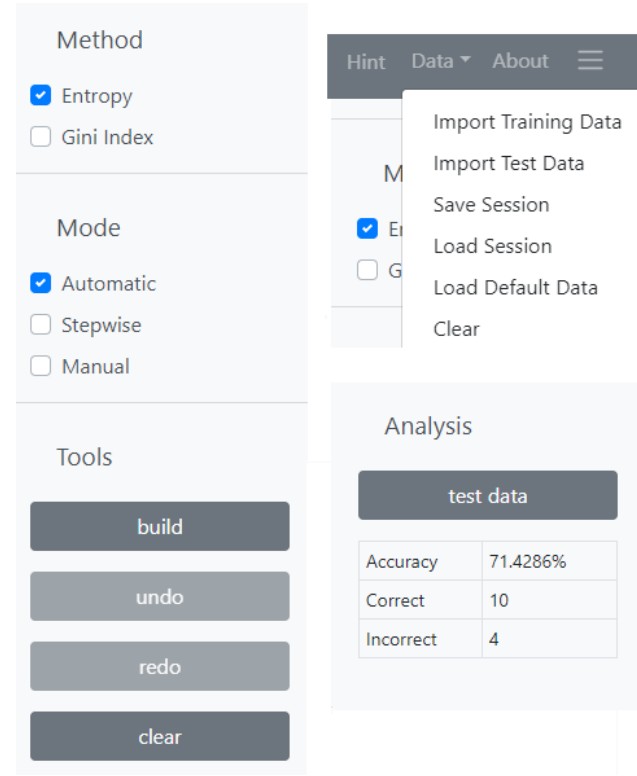

*Figure 4.* A more detailed summary of the possible settings in the web app. It is possible to load custom or default training and test data into the tool, as well as load or save complete sessions under manual mode (right). In manual mode options to *undo* and *redo* splits exist. The user can choose between the two methods Gini index or Shannon entropy to calculate the optimal DT (left).

stimulating tool for experimentation that is easy to extend and customize.

In terms of future work, the tool itself could be extended with ensemble methods, for instance. Also, noisy data handling could be included. In the current version, it has been excluded since, for beginners, it is easier to first understand the correct inner workings of the algorithm with one tree before coping with multiple of them at once. Another possible extension would be to include a means to limit the maximum tree depth and allow "impure" nodes to embrace the concept of underfitting. On the other hand, the support regarding the web app itself could be extended. As the demand of skilled ML teachers keeps growing, it would be advisable to invest in a platform that comprises as many teachers in the area as possible. Concerning the web app, all teachers could, for instance, contribute to an online platform with their respective contents and share data sets or application contexts. To accelerate AI Learning, the platform could be envisioned to cover more topics and DTs form only one section where this web app will be made available to everybody.

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
