# OpenReview forum: "An Interactive Web Application for Decision Tree Learning"
_ECMLPKDD.org/2020/Workshop/TeachML — ECML PKDD 2020 TeachML_

### Official Review · AnonReviewer1 · 2020-07-16
**Potentially useful for beginner classes, but very basic and using non-standard tree construction**

**Rating:** 6
**Confidence:** 4

**Review:**

I can see the potential of the tool introduced here to teach and explain the very basics of decision trees (and how to evaluate classifiers).

BUT:

1. I'm not sure it is entirely finished -- the "Hint" functionality did not do anything, as far as I could make out, and best practice for UI design for this kind of interactive tutorial would be to have documentation in the form of tooltips right inside the app.

2. I think the decision to use a non-binary decision tree algorithm like ID3 is suboptimal from a didactic point of view. How to find optimal *binary* splits is much easier to explain/understand and the resulting tree structures are easier to follow as well.

3. This tool only aids in the visual explanation of very simple decision tree topics, which, in my teaching experience, only rarely cause confusion or issues in the classroom, namely the basic "greedy" recursive split search and how to use the tree to classify new data.
This seems like a wasted opportunity and limits the utility of the proposed tool-- more challenging and practically very important topics that learners often struggle with in my experience, like, e.g., pruning trees to avoid overfitting, how surrogate splits work, the instability of tree structure to small data perturbations,  etc, are not covered by the functionality of the tool.

4. This tool is not based on any of the popular ML frameworks in R or Python, which seems another wasted teachingg opportunity. If it were, this would allow students to also see the code generating the respective model and thereby learn to relate theoretical concepts to the corresponding implementation features.

---

### Official Review · AnonReviewer2 · 2020-07-28
**Interesting Web Application for learning Decision Trees, but lacking some broader course context**

**Rating:** 6
**Confidence:** 5

**Review:**

This paper presents a web application for learning decision trees (referred to as DTs throughout the paper) as well as their nuances. The authors do an excellent job situating why learning decision trees is important. The web application is well explained from the students perspective, but I would have liked a few more details on the instructor's perspective.

There are a few elements of the paper that feel a bit disjointed. For example, the paper spends considerable space on the "tennis data set" but yet the default data set for the web application concerns monsters. Similar if the intended audience has a background in computer science, how does this tool use and/or enhance one's computer science training?

Additionally, I would have liked to see a connection to how this tool fits into a machine learning course. For example, is this a student's first or second contact with the material? Do students use this tool to inform pseudocode drafts?

Pros -
* Well designed tool for learning and understanding the nuances of decision trees
* Thoughtful presentation of the web application and strong motivation for learning about decision trees

Cons -
* Some issues with the structure of the current draft
* Missing explicit placement into a machine learning course

---

### Decision · Program_Chairs · 2020-07-31

**Decision:**

Accept

**Comment:**

The reviewers agree that this paper will be accepted. Thank you for your contributions.

Please register with the conference as soon as possible! See this page for details:
https://ecmlpkdd2020.net/attending/registration/.
Which asks that at least one author per paper registers until July 31, 2020.
We apologize for the very short notice.